# The Application Potential of Hop Sediments from Beer Production for Composting

**Michał Kopeć [1], Monika Mierzwa-Hersztek [1,*] , Krzysztof Gondek [1] , Katarzyna Wolny-Koładka [2] , Marek Zdaniewicz [3] and Aleksandra Suder [2]**

[1] Department of Agricultural and Environmental Chemistry, University of Agriculture in Krakow, al. Mickiewicza 21, 31-120 Krakow, Poland; michal.kopec@urk.edu.pl (M.K.); rrgondek@cyf-kr.edu.pl (K.G.)

[2] Department of Microbiology and Biomonitoring, University of Agriculture in Krakow, al. Mickiewicza 24/28, 30-059 Krakow, Poland; katarzyna.wolny@urk.edu.pl (K.W.-K.); aleksandra.suder@student.urk.edu.pl (A.S.)

[3] Department of Fermentation Technology and Microbiology, University of Agriculture in Krakow, ul. Balicka 122, 30-149 Krakow, Poland; m.zdaniewicz@urk.edu.pl

\* Correspondence: monika6_mierzwa@wp.pl

**Abstract:** The chemical composition of hop sediments from beer brewing and fermentation gives them the potential for further use. These wastes are not generated in large amounts, but the absence of proper characteristics may lead to processing errors. This study examines the possibility of using hop waste for aerobic biological transformation processes (composting). The study was carried out on two hop sediments from two different technological stages of beer production: hot trub and spent hops. Chemical, microbiological, and biochemical analyses were performed in the composted hop sediments, as was the assessment of phytotoxicity to *Lepidium sativum* L. The tested feedstocks were partially inhabited by microorganisms and thus safe from an epidemiological point of view, and they were not a source of microbial contamination. Inhibitory properties for plant development were found for hot trub, which most likely result from the organic compound content. If it is only a small portion of its biomass, the mineral composition of hot trub does not exclude the possibility of its composting. Spent hops were characterized by a significant total nitrogen content, which affected the composting process. Composting this sediment required the selection of substrates with a widely C:N ratio.

**Keywords:** hop sediments; respiratory activity; compost; brewing; phytotoxicity

## 1. Introduction

Beer production generates at least two types of solid waste associated with the hops used. These are hot trub and spent hops resulting from dry hopping [1–5]. The estimated amount of filtration sediments generated by the brewing industry (waste code 020780) ranges between 0.55 and 2.61 kg·hL$^{-1}$ of beer and totals 1.22 kg·hL$^{-1}$ on average. Filtration sediments are considered to be the largest source of waste in a brewery (17.6 kg·hL$^{-1}$), as on average they amount to 7% of the brewer's grain mass [6]. It may seem that the amount of waste generated is not that large, but when considering the extent of beer consumption in Poland by one individual (1 hL·year$^{-1}$), economic and environmental problems are very likely [4,7]. Streczyńska et al. [5] state that the value of solid wastes generated by 1 m$^3$ of a finished beer product is much smaller than that reported a decade ago, and is now equal to 51.2 kg, including filtration sediments [6].

During the brewing of the hopped wort, some of the hop components (alpha acids) change their form from insoluble to soluble (iso-alpha acids). This favors the formation of beer bitterness and reduction of the nitrogen content in the wort [2]. Waste is usually separated by centrifuging or filtering the wort or through other processes [4]. The resulting waste differs in chemical composition, which is determined by the type of feedstock, the environment pH, and the time of brewing. Waste is very hydrated (water constitutes 85%

of its volume) and, after drying, its components are on average as follows: carbon 50%, nitrogen 8%, and ash 2–5% [4,8]. As in the sediment, the concentration of both micro- and macro-elements in beer wort is not constant—it depends on the feedstock type, quantity, and quality [4]. Some studies [1] confirm that, due to the relatively high (but optimal for yeast) zinc content, the addition of hop sediment may support the fermentation process. However, it is necessary to separate the sediment and clarify the beer wort because of their importance for the rest of the process.

At a later stage, to provide a fresh hop aroma in the beer fermentation or aging process, so-called dry hopping is performed. The chemical composition of spent hops depends largely on the variety of hops used and their dosage form (cone, granules, etc.). Alpha acids in two forms were found in spent hops: non-isomerized forms (whose concentration was lower than the initial value) and iso-alpha forms. This is explained by the leaching of hop acids during long hours of exposure to the fermenting wort, and by the adsorption of the isomerized compounds found in the hopped wort on the surface of the dosed hop [3].

The application of hop sediments from the brewery is not very popular, mainly because of their bitter compounds, such as hops, which prevent their use in animal nutrition [7]. On the other hand, the literature gives examples of the use of hop sediments as a feed, but only as a 1–2% additive to brewers grains [6]. A high potential for the re-use of spent hops in the brewing process was found due to the content of $\alpha$-acids and polyphenols. Hop residue may also be successfully used to produce natural, inexpensive repellents to fight pests on stored food. Given its high content of essential oils (mostly sesquiterpenes), Bedini et al. [9] suggested the use of this byproduct (low-cost feedstock) as an ecological alternative to synthetic repellents, protecting stored food products against insects. Many substances identified in this type of waste are considered semiochemicals. The worst solution for disposing of sediments from the brewery (which was still popular 10 years ago in Poland) is to direct them to the municipal sewage system. This increases the costs of wastewater treatment and is irrational from an ecological and economic point of view [6]. In the literature, the importance of sediments as a pharmaceutical and cosmetological raw material is noted [4,10].

Nevertheless, the recovery of waste through aerobic treatment seems to be the least problematic [6,7]. Changes in organic matter resulting from composting make it possible to obtain a product that is safe for the environment. The process can be further optimized and controlled by selecting substrates that will be appropriate for the recognized chemical composition and biological activity of hop waste. This study was aimed at determining the potential use of hop waste from various stages of beer production in the composting process.

## 2. Materials and Methods

### 2.1. Feedstocks

The study was carried out on samples of hop sediments and spent hops of Polish hop varieties obtained from an industrial brewery in accordance with the course of the technological process. Hot trub (HT) was the first material analyzed, followed by spent hops (SH). The analysis was aimed at determining the microbiological and chemical composition of hop sediments. The composted materials were further analyzed for respiratory activity and phytotoxicity to *Lepidium sativum* L. Composting on a laboratory scale was performed to verify the natural usefulness of hop sediments.

### 2.2. Composting Conditions—Experimental Set-Up

The composting process was carried out under controlled conditions, and lasted 90 days. The experiment comprised 3 treatments and the samples were run in triplicate:

- I—Control (K; maize straw);
- II—Hot trub (HT) + maize straw (K);
- III—Spent hops (SH) + maize straw (K).

Hop sediments sampled from the brewery were mixed with maize straw in a 1:1 ratio calculated on a dry matter basis to obtain 2 kg of fresh mass. The maize substrate was selected because its chemical composition and C:N ratio were optimal for the process. The latter parameter changed after introducing sediments. The containers with the material were placed in a water bath and the following temperature regime was established for composting: the first 10 days: 60 °C; 11th–30th day: 50 °C, and 31st–90th day: 30 °C. In the composting period, the material was sampled 3 times every 30 days, then dried and analyzed. During composting, the material was aerated automatically for 8 cycles a day, every 3 h for 15 min (capacity of 600 L/h) and manually every second day. For the first 45 days of the experiment, the material moisture was maintained at 65%, and in the second half of the process, it was allowed to reach 40%.

### 2.3. Chemical Analysis

After composting, materials were sampled and dried at 105 °C for 24 h to determine their dry matter content by weight [11]. Samples of starting materials and composts were ashed in a chamber furnace at 450 °C for 12 h to determine their chemical composition. Subsequently, the ash was digested in diluted (1:2) $HNO_3$ and the samples were put into flasks. Redistilled water was added to adjust the final sample volume to 50 mL. The ICP-OES method was used to determine the contents of studied elements in the resulting solutions. The following parameters were determined in the suspension of compost and water (1:10 ratio): pH potentiometrically (pH meter CP-505, Elmentron, Zabrze, Poland); electrical conductivity (EC) using the conductometer CCO-501 (Elmentron, Zabrze, Poland); total carbon and nitrogen contents with the use of a Vario MAX Cube analyzer equipped with an IR sensor (Vario MAX Cube, Elementar Analysensysteme, GmbH, Germany).

### 2.4. Microbiological Analyses

The applied serial dilution method designed by Koch [12,13] with numerous microbiological substrates allowed the establishment of the following groups of microorganisms in sediments: endospore and vegetative bacteria (Trypticasein Soy Lab Agar, Biocorp, Poland, grown at 37 °C for 24 h), actinomycetes (Actinomycete Isolation Lab Agar, Biocorp, Poland, grown at 28 °C for 7 days), mold fungi (Malt Extract Agar, Biocorp, Poland, grown at 28 °C for 5 days). Additionally, potentially pathogenic bacteria were evaluated: *Staphylococcus* spp. (MSA agar, Biocorp, Poland, grown at 37 °C for 24 h), *Escherichia coli* (TBX agar, Biocorp, Poland, grown at 44 °C for 24 h), *Enterococcus faecalis* (SB agar, Biocorp, Poland, grown at 37 °C for 48 h), *Salmonella* spp. and *Shigella* spp. (SS agar, Biocorp, Poland, grown at 37 °C for 24 h), *Clostridium perfringens* (SC agar, Biocorp, Poland, grown at 37 °C for 24 h). From an epidemiological point of view, the presence of these bacteria may compromise health and safety and is an important indicator of microbial contamination [14,15]. The dilution culture method was applied to determine the number of colony forming units (CFU) of microorganisms. The obtained values were converted into 1 g D.M. of the tested feedstock [16].

### 2.5. Respiratory Activity

Respiratory activity (the effect of $CO_2$ changes in the closed vessel volume) of the starting and composted materials was determined by a manometric method, using an OxiTop measuring apparatus, in accordance with ISO 14855-1:2005 [17] (Figure 1). Samples used in the study had 40 g of fresh mass. The resulting samples' dry matter content was determined after drying at 105 °C for 12 h [18].

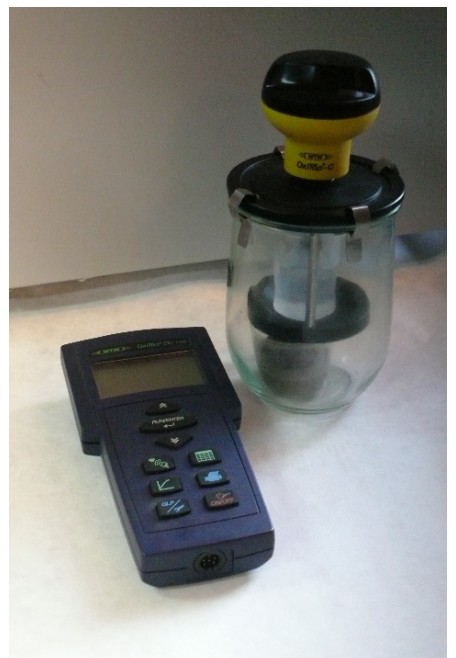
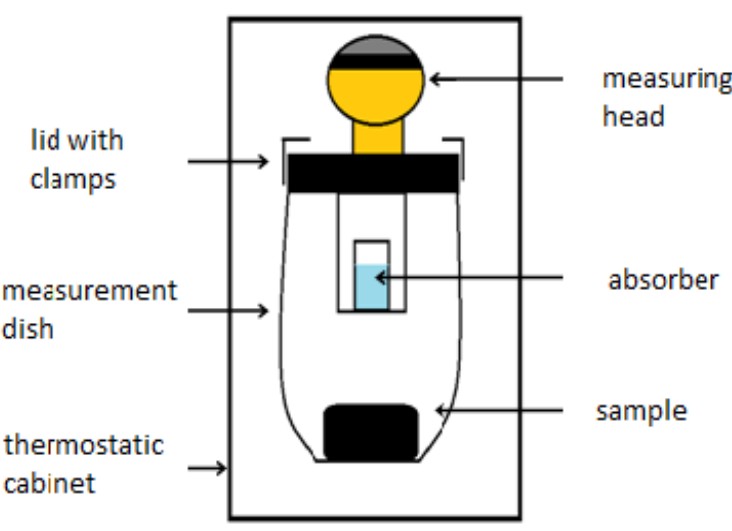

**Figure 1.** OxiTop® Control measuring set for respiratory activity.

The manometric measurement of respiratory activity of the studied materials involved the recording of pressure changes in closed containers in a continuous system (Figure 1). Pressure changes are proportional to the amount of oxygen consumed by the sample as a result of respiratory processes occurring within it [19,20]. The respiratory activity of each material was measured at different times (including a lag-phase period to stabilize conditions and the period of actual parameter measurements). The automatic recording of pressure changes took place every 30 min. The obtained equivalent $CO_2$ quantities were absorbed by 1 mol·dm$^{-3}$ NaOH contained in the vessels. Respiratory activity was measured with the use of an installation of 1.0 dm$^3$ measuring bottles with accessories, which were put into a thermostatic cabinet with a constant temperature of 25 °C ($\pm$0.1 °C) maintained. The data obtained were sent to the controller via an infrared interface and then to a PC with the use of the Achat OC program. The following formula was used to convert the respiratory activity values into dry matter data (1):

$$BA = \frac{M_{O2}}{R \times T} \times \frac{V_{fr}}{m_{Bt}} \times |\Delta p| \ (\text{mg} \times O_2 \times (\text{g} \times \text{h})^{-1}) \tag{1}$$

where $BA$ is biological activity, $M_{O2}$ is the molecular weight of oxygen (31,998 mg $\times$ mol$^{-1}$), $R$ is the universal gas constant (83.14 L·hPa (K·mol$^{-1}$)$^{-1}$, $T$ is the measurement temperature (K), $m_{Bt}$ is the dry mass weight in the composted material (kg), $|\Delta p|$ is the pressure change (hPa), and $V_{fr}$ is the free gas volume.

The growth of garden cress (*Lepidium sativum* L.) with sediment or compost extracts was analyzed. For this purpose, sediment and compost extracts (in a material:water ratio of 1:10, extraction for 24 h) were obtained in a closed vessel in which carbon dioxide was absorbed and manometric measurements performed (OxiTop® Control) [19,20]. A vacuum pump was used to 0.45 µm filter the extract. The control was in a vessel in which only redistilled water was used for the seedling growth test. The test was carried out on 0.268 g of seeds (i.e., 100 seeds of *Lepidium sativum* L.), and lasted 4 days. The results of the seedling growth tests were calculated using Equation (1) and presented in the form of regression equations of oxygen consumption 48 h after the addition of sediment extract, expressed in $O_2 \times (\text{g} \times \text{h})^{-1}$.

*2.6. Statistical Analysis*

The statistical compilation of the results of chemical and biochemical analyses involved presentation of the arithmetic mean of replicates and standard deviation. This was also achieved by carrying out variance analysis and by synthesis using Tukey's test method with the adopted significance level of $p \leq 0.05$.

## 3. Results and Discussion

*3.1. Chemical Composition of Hop Sediments before Composting*

The pH, EC values, and chemical composition of the tested sediments are presented in Table 1. A lower pH was determined for the hot trub. A characteristic feature of the hot trub was a higher value of electric conductivity. This probably resulted from a greater content of leached potassium. The content of the tested elements in the hot trub was generally smaller or at a similar level to the content determined in the spent hops (SH) obtained at later stages of the process. The increased total content of copper and zinc in the technological process also deserves attention. The Cu content in the sediments was significantly higher compared to the content determined in the plant material used for beer production. The total copper content in the HT was over 20 times lower, and the total zinc content over 7 times lower, than in SH. These differences between sediments may have also resulted from changes occurring at different temperatures and from the environment of the vat material.

**Table 1.** Electrical conductivity, pH, and total element content in hop sediments.

| Parameter | Unit | HT | SH |
|---|---|---|---|
| pH (1:10) | | 4.96 (0.09) | 5.19 (0.08) |
| EC (1:10) * | $mS \cdot cm^{-1}$ | 1.855 (0.06) | 1.282 (0.08) |
| N | $g \cdot kg^{-1}$ | 27.84 (0.12) | 49.96 (0.35) |
| C | $g \cdot kg^{-1}$ | 522.9 (10.5) | 490.3 (8.20) |
| C:N ratio | | 18.78 | 9.81 |
| Total Na | $g \cdot kg^{-1}$ | 0.12 (0.00) | 0.10 (0.00) |
| Total K | $g \cdot kg^{-1}$ | 9.92 (0.26) | 4.67 (0.05) |
| Total Mg | $g \cdot kg^{-1}$ | 2.41 (0.03) | 4.39 (0.01) |
| Total Ca | $g \cdot kg^{-1}$ | 8.04 (0.05) | 9.08 (0.01) |
| Total P | $g \cdot kg^{-1}$ | 3.36 (0.12) | 9.36 (0.05) |
| Total Cd | $mg \cdot kg^{-1}$ | 0.03 (0.00) | 0.10 (0.00) |
| Total Cr | $mg \cdot kg^{-1}$ | 0.68 (0.07) | 0.64 (0.09) |
| Total Cu | $mg \cdot kg^{-1}$ | 5.74 (0.05) | 124.88 (2.01) |
| Total Fe | $mg \cdot kg^{-1}$ | 314.75 (9.55) | 307.25 (17.00) |
| Total Mn | $mg \cdot kg^{-1}$ | 37.68 (0.57) | 48.34 (0.05) |
| Total Ni | $mg \cdot kg^{-1}$ | 1.10 (0.00) | 0.44 (0.02) |
| Total Pb | $mg \cdot kg^{-1}$ | 0.45 (0.00) | 0.53 (0.07) |
| Total Zn | $mg \cdot kg^{-1}$ | 30.13 (0.21) | 216.48 (0.78) |

* EC: electrical conductivity; standard deviation is given in brackets; *n* = 3.

In the case of zinc, this was probably due to the supplementation of the fermentation process with ions of this element, which promotes the development of yeast [21]. Poreda et al. [21] state that the rate of $Zn^{2+}$ extraction from malt is very low and 95% of Zn ions from feedstocks are removed with waste. The procedure modification proposed by the authors, consisting of the addition of $Zn^{2+}$ salt only after a significant volume of clarified wort is removed from the whirlpool tank, retained almost 90% of the added $Zn^{2+}$ ions.

When it comes to the biological processing of sediments, total nitrogen and carbon contents and their proportions are important in composting and biogasification technologies. Spent hops have a high nitrogen content and a low C:N ratio, which, in the composting process, suggests the need to supplement the material with a higher carbon content in relation to nitrogen. For hot trub, the C:N ratio is also incorrect, but its improvement theoretically requires a smaller material portion.

### 3.2. Microbiological Analysis

The microbiological analysis of hop sediments excluded the presence of virtually all the microorganisms tested for. Only vegetative bacteria were found in low numbers in the tested materials: 1387 CFU·g$^{-1}$ D.M. (hot trub (HT)), 223,800 CFU·g$^{-1}$ D.M. (spent hops (SH)). After 20 days of incubation, these materials respectively had counts of 1143 CFU·g$^{-1}$ D.M. and 150,767 CFU·g$^{-1}$ D.M. Based on these results, it can be concluded that the feedstocks tested are safe from an epidemiological point of view. These properties will influence the subsequent use of the hop sludge [22,23].

### 3.3. Respiratory Activity of Sediments before Composting

Under aerobic conditions, hydrated organic material will show varying respiratory activity resulting from microbial activity. The analyzed materials (HT and SH) had extremely different oxygen demands during incubation (Figure 2).

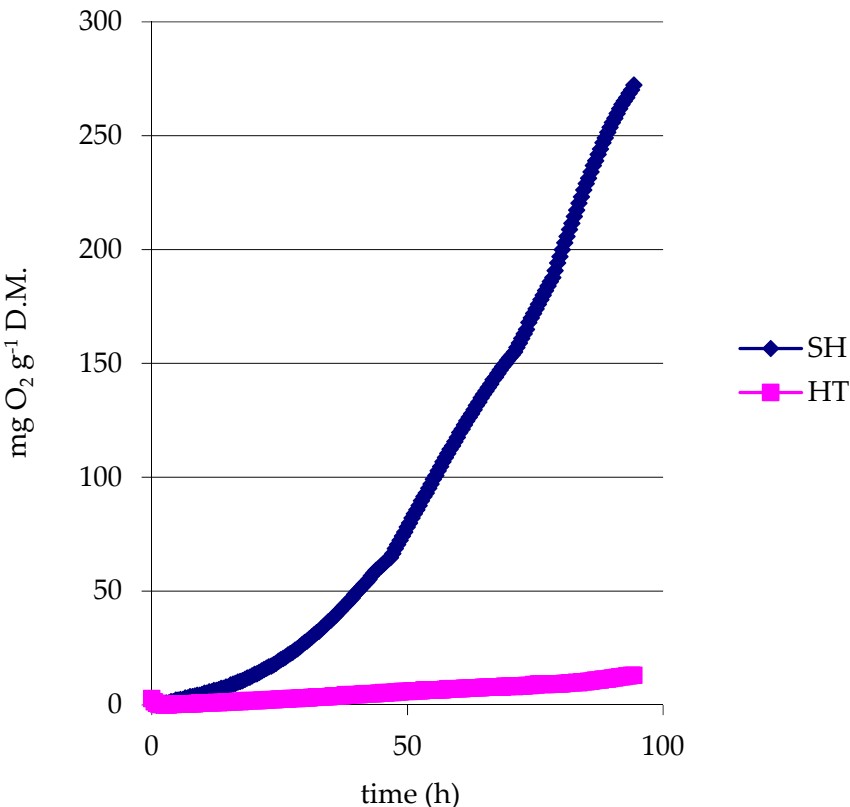

**Figure 2.** Experimental values (*n* = 360) showing changes in the respiratory activity of hops over a period of 96 h.

To assess respiratory activity, a regression straight line was drawn after rejection of the lag-phase time (36 h):

- $y(HT) = 0.136x - 0.9672$ $R^2 = 0.9711$;
- $y(SH) = 4.082x - 123.51$ $R^2 = 0.9887$.

The values of the equation's directional coefficients differ from each other by 30 times, which confirms the high respiratory activity of spent hops and the inhibition of biochemical processes in the incubated hot trub. This is confirmed by microbiological tests on sediments.

In the case of filtration sediments from the brewery, the calculated oxygen demand is most likely due to the formation of $CO_2$ caused by the lysis of dead cells. This makes the proposed method unsuitable for the interpretation of direct respiratory activity; therefore, the phytotoxicity assessment was adopted for further characterization of the sediments.

In order to verify the plant's response to direct contact with sediment materials, a biological test was carried out based on the growth of garden cress in contact with aqueous sediment extract. The oxygen demand curves of germinating garden cress seedlings varied in the middle of the second day of plant growth (Figure 3). Both extracts were growth inhibitors compared to water; however, after the second day, the SH extract was not phytotoxic. Plant germination was inhibited in contact with HT extract, and after 4 days only single germinated seeds were found.

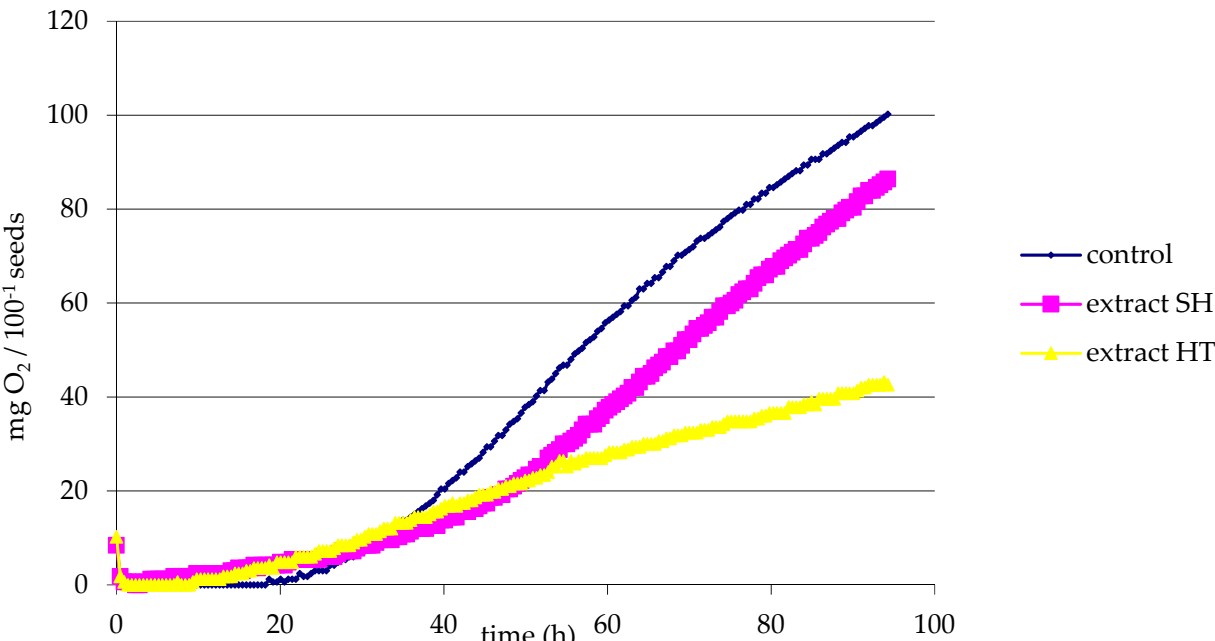

**Figure 3.** Experimental values (*n* = 360) showing changes in oxygen demand over 96 h of garden cress growth.

The effect of extracts was assessed on the basis of oxygen demand regression straight lines drawn for garden cress for the third and fourth day of measurements:

- $y(control) = 1.414x - 29.725$ $R^2 = 0.9899$;
- $y(HT) = 0.454x + 0.286$ $R^2 = 0.9942$;
- $y(SH) = 1.458x - 49.967$ $R^2 = 0.9993$.

The values of the equation's directional coefficients for treatments with seeds in contact with water or spent hops extracts are similar (1.414 and 1.458, respectively), which indicates that there were no differences in garden cress growth. The directional coefficient value of the equation describing the oxygen demand in the treatment with HT extract was one-third the value of the other two, which confirms a significant inhibitory effect on garden cress growth.

Slight differences in mineral composition and pH were found between the extracts (Table 2). It is worth noting the inverted ratio of monovalent to divalent cations and the greater share of hydronium ions in the HT extract. However, the effect of unanalyzed organic components cannot be excluded.

**Table 2.** Mineral composition of extracts used for the germination test.

| Parameter | Unit | Control | HT Extract | SH Extract |
|---|---|---|---|---|
| pH | - | 6.27 (0.09) ** | 4.70 (0.21) | 5.64 (0.11) |
| EC | $mS \cdot cm^{-1}$ | 0.062 (0.002) | 0.478 (0.010) | 0.275 (0.020) |
| Na | $\mu g \cdot cm^{-3}$ | 0.575 (0.021) | 3.591 (0.054) | 2.928 (0.092) |
| K | $\mu g \cdot cm^{-3}$ | 8.321 (0.321) | 134.4 (0.028) | 51.89 (0.120) |
| Mg | $\mu g \cdot cm^{-3}$ | 0.208 (0.011) | 16.44 (0.087) | 36.93 (0.890) |
| Ca | $\mu g \cdot cm^{-3}$ | 2.634 (0.121) | 18.93 (0.08) | 37.25 (0.43) |
| P | $\mu g \cdot cm^{-3}$ | 1.215 (0.012) | 37.16 (0.07) | 89.62 (1.23) |
| Cd | $\mu g \cdot cm^{-3}$ | 0.001 (0.001) | 0.001 (0.001) | 0.001 (0.001) |
| Cr | $\mu g \cdot cm^{-3}$ | 0.001 (0.001) | 0.005 (0.001) | 0.004 (0.001) |
| Cu | $\mu g \cdot cm^{-3}$ | nd * | 0.008 (0.001) | 0.057 (0.009) |
| Fe | $\mu g \cdot cm^{-3}$ | 0.010 (0.001) | 0.109 (0.001) | 0.083 (0.003) |
| Mn | $\mu g \cdot cm^{-3}$ | 0.072 (0.002) | 0.183 (0.002) | 0.252 (0.019) |
| Ni | $\mu g \cdot cm^{-3}$ | 0.002 (0.001) | 0.001 (0.001) | 0.001 (0.001) |
| Pb | $\mu g \cdot cm^{-3}$ | 0.004 (0.001) | 0.005 (0.001) | 0.008 (0.001) |
| Zn | $\mu g \cdot cm^{-3}$ | 0.106 (0.003) | 0.124 (0.009) | 0.177 (0.019) |

* nd: not determined; ** standard deviation is given in brackets; $n = 3$.

### 3.4. Respiratory Activity of the Composted Hop Sediments

The composting process was carried out on hop sediments mixed with maize straw. The largest changes involving carbon oxidation took place within the first 30 days of composting (Table 3). They were statistically significant between treatments. After another 30 days, carbon losses were even higher, although they were insignificant in the case of maize and hot trub. The lack of significant differences in the dry matter residue between the 60th and 90th day indicates the inhibition of respiratory processes in all treatments. This was confirmed by the measurement of respiratory activity between the 85th and 90th day, which corresponds to the AT4 parameter (respiratory activity over 4 days = 96 h). The trend lines of equations and their solutions are as follows:

- $y(K) = 0.0002x + 0.0023$ for x = 96 y = 0.0215 mg $O_2 \cdot g^{-1}$ D.M.;
- $y(K + HT) = (4 \times 10^5)x + 0.0015$ for x = 96 y = 0.00534 mg $O_2 \cdot g^{-1}$ D.M.;
- $y(K + SH) = (8 \times 10^{-5})x + 0.0024$ for x = 96 y = 0.01008 mg $O_2 \cdot g^{-1}$ D.M., where K is maize straw.

**Table 3.** Relative changes in dry matter residue (% relative to starting material) during the composting process.

| Treatment | Day of Composting | | | |
|---|---|---|---|---|
| | 0 | 30 | 60 | 90 |
| K | 100 [a],* | 32.14 [f] | 23.36 [g] | 22.77 [g] |
| K + HT | 100 [a] | 56.89 [b] | 55.61 [b,c] | 53.62 [c] |
| K + SH | 100 [a] | 48.26 [d] | 41.67 [e] | 40.65 [e] |

* Each value represents the mean of three replicates; different letters within a column indicate a significant difference at $p \leq 0.05$ according to Tukey's HSD test; factor: time × sediment.

AT4 values are extremely small, but they confirm the relationship between the measurements of substrates (hop sediments) made before composting. The major changes that occurred in the organic material during composting related to the carbon:nitrogen ratio (Table 4). After composting, carbon loss caused higher nitrogen concentrations. The largest changes were observed after composting maize.

**Table 4.** Contents of nitrogen and carbon ($g \cdot kg^{-1}$) during composting (on start day, and after day 30, 60, and 90).

| Parameter | Process Day | K | K + HT | K + SH |
|---|---|---|---|---|
| Total N | 0 | 10.5 [a] | 18.4 [a] | 30.4 [b] |
| | 30 | 31.2 [b] | 33.0 [b] | 45.1 [c,d] |
| | 60 | 36.3 [b,c] | 32.2 [b] | 49.5 [d] |
| | 90 | 37.6 [b,c] | 33.8 [b] | 52.0 [d] |
| Total C | 0 | 429.8 [a,b] | 513.8 [e] | 475.3 [b,c,d,e] |
| | 30 | 426.4 [a,b,c] | 494.3 [e] | 462.5 [c,d,e] |
| | 60 | 406.7 [a,b] | 483.2 [d,e] | 458.0 [c,d,e] |
| | 90 | 386.3 [a] | 482.6 [d,e] | 434.9 [a,b,c,d] |
| C:N ratio | 0 | 40.9 [e] | 27.9 [d] | 15.6 [c] |
| | 30 | 13.6 [b,c] | 15.0 [c] | 10.2 [a,b,c] |
| | 60 | 11.2 [a,b,c] | 14.9 [c] | 9.3 [a,b] |
| | 90 | 10.3 [a,b,c] | 14.3 [b,c] | 8.4 [a] |

Each value represents the mean of three replicates ± standard deviation. Different letters within a column indicate a significant difference at $p \leq 0.05$ according to Tukey's HSD test; factor: time × sediment.

The adopted maize:hop sediment ratio in the composted mixtures did not allow them to achieve the required C:N ratio, which was below 30. The high nitrogen content in sediments reduced the C:N ratio to 27.9 for HT and 15.6 for SH. Unfortunately, these values are considered incorrect for composting [23]. A structural material with a much higher C:N ratio is required for composting hop sediments, which are rich in nitrogen. This could be straw or sawdust, for instance. A significant decrease in the C:N ratio during composting was reported in all treatments in relation to the starting material. The largest C:N ratio reduction was recorded in the first composting period (up to day 30). Subsequent changes in all treatments had no statistical significance.

## 4. Conclusions

1. Hop sediments produced at various stages of the brewing process differ significantly in chemical and biological reactivity, which affects their possible biological processing.
2. Hop sediments were free from microbiological contamination that may pose an epidemiological threat.
3. The SH sediment was very rich in total nitrogen (50 g/kg D.M.), which affected the composting process. Composting this sediment required the selection of substrates with a wide C:N ratio. However, the sediment is suitable for biological processing.
4. In the case of HT, plant growth inhibitory properties were found both before and after composting, which were most likely not due to the mineral composition. Organic compounds and the sterilization of the substrate as a result of wort boiling may be inhibitors of biological changes.
5. Stable compost with hop sediments could be obtained after 60 days of an intensive aerobic transformation, with the appropriate quantitative and qualitative selection of substrates. In the last 4 days of the experiment, the oxygen demand was very low and amounted to below 0.0215 mg $O_2$ $g^{-1}$ D.M.

**Author Contributions:** Conceptualization: M.K., M.M.-H., M.Z., and K.G.; methodology: M.K., M.M.-H., K.W.-K., and K.G.; software: M.K.; validation: M.K., M.M.-H., and K.G.; formal analysis: M.K.; investigation: M.K., M.M.-H., M.Z., K.W.-K., and K.G.; resources: M.K.; data curation: M.K., M.M.-H., K.W.-K., and K.G.; writing—original draft preparation: M.K.; writing—review and editing: M.K., M.M.-H., M.Z., K.W.-K., K.G., and A.S.; visualization: M.K.; supervision: M.K., M.M.-H., M.Z., K.W.-K., and K.G.; project administration: M.Z.; funding acquisition: M.Z. All authors have read and agreed to the published version of the manuscript.

**Funding:** This work was financially supported by Grant LIDER 46/0185/L-9/17/NCBR/2018.

**Institutional Review Board Statement:** Not applicable.

**Informed Consent Statement:** Not applicable.

**Data Availability Statement:** Not applicable.

**Conflicts of Interest:** The authors declare no conflict of interest.

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
