# Peer review of "The Application Potential of Hop Sediments from Beer Production for Composting"

_sustainability, doi:10.3390/su13116409_

Round 1
Reviewer 1 Report
Comments:
- Line 37 - references [6] – is it correctly quoted?; in text is “ grains mass.[6].” – it should be grain mass [6].
- Materials an methods:
- Lines 118-122 - is total carbon and nitrogen were determined in the suspension of compost and water - if so, please explain why?
- Results and discussion
- “3.1.Chemical composition of hop sediments before and after composting” - Is the subsection title correct? The content shows that the chemical properties of hop sediments before composting are discussed.
- “3.4. Respiratory activity of the composted hop sediments” - Is the subsection title correct? In lines 236-267 the authors describe relative changes in the dry matter residue and in lines 285-300 changes of N and C contents and the C: N ratio.
- Lines 264-265 – in text is: “After 30 days, carbon losses were even higher” . It should be after another 30 days or after 60 days.
- Lines 267-269 – “This was confirmed by the measurement of respiratory activity between the 85th and 90th day, which corresponds to the AT4 parameter (respiratory activity over 4 days = 96 hours)” - The paper does not include the results of compost respiratory activity between the 85th and 90th day.
- The same results are shown in table 3 and figure 4. Is it necessary?
- The description of the chemical analyzes (verses 121-122) shows that the total carbon and nitrogen content was determined. Such a significant increase in total nitrogen content during composting requires some explanation
- Lines 292-296: “The adopted maize : hop sediment ratio in the composted mixtures did not allow to achieve the required C : N ratio, which was below 30. The high nitrogen content in sediments reduced the C : N ratio to 27.9 for HT and 15.6 for SH. Unfortunately, these values are considered incorrect for composting. A structural material with a much higher C : N ratio is required for composting hop sediments, which are rich in nitrogen” – So why, at the planning stage of the experiment, the ratio of the maize straw to the hop sediments was not established so that the C: N ratio was optimal for the composted materials?
- In text is: „4. . Conclusions” – should be: 4. Conclusions
Author Response
Dear Editor,
I am pleased to resubmit for publication the revised version of the manuscript: " The application potential of hop sediments from beer produc-tion for composting" (Manuscript ID sustainability-1196467) after taking into account Reviewers.
Firstly, we are thankful to the Reviewers and Editor for their valuable time and effort to review our manuscript and appreciate the reviewer’s comments that would certainly improve the quality of the manuscript. The responses to the comments are provided point by point as raised by the reviewers. The revisions made were highlighted in yellow color for the easy reference. Additionally other minor corrections/revisions were made in the manuscripts that were also highlighted in yellow color. We are sure that this would satisfy reviewer’s concerns. Please find below the responses to the corrections point by point as raised by the reviewers, along with the list of changes that we have made in the revised manuscript.
|
Yes |
|
|
|
· Lines 118-122 - is total carbon and nitrogen were determined in the suspension of compost and water - if so, please explain why? |
Corrected in the paper – I accept the remark |
|
|
|
|
|
|
· “3.1.Chemical composition of hop sediments before and after composting” - Is the subsection title correct? The content shows that the chemical properties of hop sediments before composting are discussed. |
Corrected in the paper – I accept the remark however it was before composting (not after) |
|
· “3.4. Respiratory activity of the composted hop sediments” - Is the subsection title correct? In lines 236-267 the authors describe relative changes in the dry matter residue and in lines 285-300 changes of N and C contents and the C: N ratio. |
Corrected in the paper |
|
· Lines 264-265 – in text is: “After 30 days, carbon losses were even higher” . It should be after another 30 days or after 60 days. |
The correction was included in the paper |
|
· Lines 267-269 – “This was confirmed by the measurement of respiratory activity between the 85th and 90th day, which corresponds to the AT4 parameter (respiratory activity over 4 days = 96 hours)” - The paper does not include the results of compost respiratory activity between the 85th and 90th day. |
It is correct. The trend lines below refer to the information |
|
· The same results are shown in table 3 and figure 4. Is it necessary? |
The graph was deleted |
|
· The description of the chemical analyzes (verses 121-122) shows that the total carbon and nitrogen content was determined. Such a significant increase in total nitrogen content during composting requires some explanation |
Changes in N and C are confirmed by changes in dry weight, under composting conditions, N increases due to carbon oxidation |
|
· Lines 292-296: “The adopted maize : hop sediment ratio in the composted mixtures did not allow to achieve the required C : N ratio, which was below 30. The high nitrogen content in sediments reduced the C : N ratio to 27.9 for HT and 15.6 for SH. Unfortunately, these values are considered incorrect for composting. A structural material with a much higher C : N ratio is required for composting hop sediments, which are rich in nitrogen” – So why, at the planning stage of the experiment, the ratio of the maize straw to the hop sediments was not established so that the C: N ratio was optimal for the composted materials? |
At the beginning, an attempt was made to carry out the process under the same conditions (also taking into account the masses) for both materials, hence the inconvenience but the possibility of comparing |
|
· In text is: „4. . Conclusions” – should be: 4. . Conclusions |
Corrected |
The authors would like to thank the reviewers for the time and effort to read the manuscript and indicate its shortcomings. The authors have made every effort to improve the quality of the paper and hope that its current form will be appropriate and accepted.
I look forward to hearing from you at your earliest convenience.
With best regards,
Monika Mierzwa-Hersztek

Reviewer 2 Report
Manuscript number: sustainability-1196467
Article Type: Research Article
Comments to Authors: Michał Kopeć, Monika Mierzwa-Hersztek, Krzysztof Gondek, Katarzyna Wolny-Koładka, Marek Zdaniewicz and Aleksandra Suder
Title: The application potential of hop sediments from beer production for composting
The Authors investigated potential use of hop waste from various stages of beer production in the composting process. Rich in total nitrogen the SH sediment, affected the composting process. Composting this sediment required selection of substrates with a wide C:N ratio. The HT had inhibitory properties and were found for plant growth, both before and after composting.
Comments
Chapter 2.3. please indicate the literature reference for the dry matter determination method
Chapter 2.4: the authors put information about method designed by Koch, but they do not provide literature in the methodology and in the bibliography list
Chapter 3: in the results and discussion chapter, please also discuss the obtained results with the literature, there was only one literature item in the discussion
Figure 1: in my opinion the figure 1 is not necessary
Figure 2 and 3: the presented charts are not graphically acceptable, please replace them with lines, e.g. with different markers
Lines 271-273: please correct O2
Figure 3: please present the axes from the zero point, not minus 20
Table 2: please use superscripts in units (cm-1) and in the standard deviation use three places after comma in every value
Table 3 and Figure 4 show the same results, but in different way (numbers and graphs); this is duplicate results; please present either in the table or in the graph
Table 3 and 4: below the table there is information that there are standard deviation values in the table, but they are not
Conclusions: please support conclusions by the results

Author Response
Dear Editor,
I am pleased to resubmit for publication the revised version of the manuscript: " The application potential of hop sediments from beer produc-tion for composting" (Manuscript ID sustainability-1196467) after taking into account Reviewers.
Firstly, we are thankful to the Reviewers and Editor for their valuable time and effort to review our manuscript and appreciate the reviewer’s comments that would certainly improve the quality of the manuscript. The responses to the comments are provided point by point as raised by the reviewers. The revisions made were highlighted in yellow color for the easy reference. Additionally other minor corrections/revisions were made in the manuscripts that were also highlighted in yellow color. We are sure that this would satisfy reviewer’s concerns. Please find below the responses to the corrections point by point as raised by the reviewers, along with the list of changes that we have made in the revised manuscript.
|
Chapter 2.3. please indicate the literature reference for the dry matter determination method |
Done |
|
Chapter 2.4: the authors put information about method designed by Koch, but they do not provide literature in the methodology and in the bibliography list |
2 items of the bibliography were supplemented |
|
Chapter 3: in the results and discussion chapter, please also discuss the obtained results with the literature, there was only one literature item in the discussion |
3 items of the bibliography were supplemented |
|
Figure 1: in my opinion the figure 1 is not necessary |
I would rather leave it for the sake of presenting a method that is little known in the context of manometric measurement manometrycznego
|
|
Figure 2 and 3: the presented charts are not graphically acceptable, please replace them with lines, e.g. with different markers |
Figures description has been supplemented, these are experimental data, not a model
|
|
Lines 271-273: please correct O2 |
Corrected |
|
Figure 3: please present the axes from the zero point, not minus 20 |
Done |
|
Table 2: please use superscripts in units (cm-1) and in the standard deviation use three places after comma in every value |
Indexes were fixed. However the use of three places after comma in every value is statistically unjustified
|
|
Table 3 and Figure 4 show the same results, but in different way (numbers and graphs); this is duplicate results; please present either in the table or in the graph |
Graph 4 has been deleted
|
|
Table 3 and 4: below the table there is information that there are standard deviation values in the table, but they are not |
Removed from the description
|
|
Conclusions: please support conclusions by the results |
The authors believe that methodologically, the conclusions should not contain results, but an attempt to generalize them; however, 2 results were entered |
The authors would like to thank the reviewers for the time and effort to read the manuscript and indicate its shortcomings. The authors have made every effort to improve the quality of the paper and hope that its current form will be appropriate and accepted.
I look forward to hearing from you at your earliest convenience.
With best regards,
Monika Mierzwa-Hersztek

Round 2
Reviewer 2 Report
I accept in present form